# Spanish Cross-Cultural Adaptation of the Australian Pelvic Floor Questionnaire

**DOI:** 10.3390/jpm13060940

**Published:** 2023-06-01

**Authors:** Esther M. Medrano-Sánchez, Ana Pérez-Carricondo, Pilar Beteta-Romero, Esther Díaz-Mohedo

**Affiliations:** 1Department of Physical Therapy, University of Seville, Avenzoar St., 41009 Seville, Spain; 2Physical Therapist, Rue Edmond Faulat, 33440 Ambares et Lagrave, France; 3Physical Therapist, Enrique Enriquez St., 18800 Baza, Spain; 4Department of Physical Therapy, University of Málaga, Francisco Peñalosa Av, 29071 Málaga, Spain

**Keywords:** pelvic floor disorders, validation study

## Abstract

The main objectives of this study were to carry out the translation and cross-cultural adaptation of the Australian Pelvic Floor Questionnaire (APFQ) into Spanish and the evaluation of its psychometric properties of validity and reliability in the Spanish population. The APFQ was translated into Spanish and back-translated into its original language by native speakers; it was verified that there was a semantic similarity. A pilot test was carried out on a group of 10 women. The study sample was made up of 104 subjects. They were asked to fill in the APFQ twice, 15 days apart. Codes were assigned so they could link to the test and retest. The Questionnaire on Pelvic Floor Dysfunctions–short version (PFDI-20) and the Women’s Sexual Function Questionnaire (FSM) were also completed. The reliability, criterion and construct validity, and stability were studied. A Cronbach’s alpha of 0.795 was obtained from the complete questionnaire. For each dimension, Cronbach’s alpha was 0.864 for bladder function; 0.796 for bowel function; 0.851 for prolapse; and 0.418 for sexual function (0.67 with the suppression of item 37). The APFQ shows a significant correlation with PFDI-20 in urinary function (rho: 0.704, *p* = 0.000), intestinal function (rho: 0.462, *p* = 0.000), and prolapse symptoms (rho: 0.337, *p* = 0.000). The test-retest analysis showed high reproducibility. The Spanish version of the APFQ is a reliable and valid tool to assess symptoms and impacts on quality of life due to pelvic floor dysfunctions in the Spanish population. However, a review of some of its items could increase its reliability.

## 1. Introduction

Pelvic floor dysfunctions include conditions related to urinary function, defecation, and sexual function, Pelvic Organ Prolapse (POP), and pain [1], which may be associated with a wide variety of symptoms and anatomical changes in the functions of the pelvic floor musculature. According to these symptoms, we can talk about situations of hypertonicity/hypotonicity in musculature, alterations in the coordination of muscular work, and failures in the support system, all of which can cause the aforementioned dysfunctions [2]. These symptoms are even more concerning in neurological patients, with a prevalence of 80% of patients having pelvic floor disorders [3]. These conditions have a negative impact on quality of life and in different areas of the patient’s life, such as the social, psychological, work, or sexual spheres [4,5,6]. In addition, these dysfunctions are common among the female world population [7,8], and their prevalence is expected to increase due to an increase in life expectancy [9]. Pelvic floor dysfunctions include a long list of clinical pathologies such as urinary incontinence (UI), fecal incontinence (IF), POP, sexual dysfunctions, and pain syndromes in the perineal area [10].

Due to their obstetric history and anatomical characteristics, the female gender is more prone to suffer from pelvic floor disorders in comparison with men [11]. Constipation, stimulants like coffee, obesity, hypertension, sedatives, or anti-inflammatory drugs may be associated with the onset or worsening of pelvic floor dysfunction symptoms [12]. Concerning the obstetrics background, a long labor stage of delivery, vaginal delivery, pelvic floor tears or incontinence during pregnancy are risk factors for pelvic floor dysfunctions subsequently [13]. High-impact sports have also been described as another risk factor for pelvic floor disorders [14,15].

UI is one of the most prevalent pelvic floor dysfunctions among women in adulthood, estimated at between 30% and 60% [16]. Among the conservative therapeutic options to treat UI, those that are currently available include pelvic floor muscle training, the use of cones for stress urinary incontinence, and electrical stimulation for urge urinary incontinence [17]. Questionnaires are very useful tools in evaluating pelvic floor dysfunctions [18]. There are a few validated questionnaires available in Spanish that evaluate symptoms and specific conditions of the pelvic floor [19]; however, they do not provide a global vision of the patient’s status in terms of the pelvic floor since most of them focus on specific pathologies or aspects, such as the short form Urogenital Distress Inventory (UDI-6) [20] or the Pelvic Floor Dysfunctions Questionnaire-Short Form (PFDI-20) [21], which assesses urinary incontinence, or the Prolapse Quality of Life Questionnaire (P-QoL) [22], which evaluates the quality of life of patients with prolapse.

The Australian Pelvic Floor Questionnaire (APFQ) evaluates all the dimensions susceptible to pelvic floor dysfunction, thus obtaining more global and complete information on the degree of involvement of the person completing the questionnaire. This questionnaire assesses four dimensions, urinary function (questions 1–15), bowel function (questions 16–27), prolapse symptoms (questions 28–32), and sexual function (questions 33–42), as well as the patient’s perspective regarding their quality of life. Although there are dysfunctions that can be objectively demonstrated, it is essential to know the patient’s perspective in terms of severity and effects on their daily life [23].

For most of the questions, Likert-type answers with four options are used, except for questions 33, 34, and 35, which are scored differently.

To calculate the total score, the scores of the questions of each dimension are added together, divided by the number of questions from each dimension, and multiplied by 10. The maximum score for each section is 10, with a maximum achievable total score of 40. For sexually inactive women, the maximum possible score is 30. The APFQ was first validated as a questionnaire administered by a professional [24], and then it was validated as a self-administered questionnaire [25].

With this study, we aim to perform a translation, cross-cultural adaptation, and validation of the APFQ into Spanish to obtain a tool that evaluates all spheres of the pelvic floor, including urinary function, bowel function, prolapse symptoms, and sexual function.

## 2. Materials and Methods

### 2.1. Study Design, Participants, and Data Collection

A cross-sectional, observational study was designed to validate the APFQ in the Spanish population. To conduct the study, we followed the STROBE guidelines [26] and “The Checklist for Reporting Results of Internet E-Surveys” (CHERRIES) guidelines [27].

Women between the ages of 18 and 65 and native Spanish speakers who presented symptoms, signs and/or a diagnosis of pelvic floor dysfunction were selected, according to the International Urogynecological Association (IUGA) and International Continence Society (ICS) Joint Report on the Terminology for Female Pelvic Floor Dysfunction [1]. The main reason for this inclusion criteria was the availability to answer an online self-administrated questionnaire and the fact that the authors were interested in assessing sexual dysfunction without association with other pathologies of the elderly. A non-probability sampling technique was used for convenience. The participants were selected in the physiotherapy offices of the authors of this work and in the offices of collaborating physiotherapists as long as they met the inclusion criteria. In addition, the APFQ was disseminated through social networks.

The exclusion criteria were not accepting the informed consent and the incorrect or incomplete completion of the survey, which was carried out online, guaranteeing the anonymity of the participants. As general data, age, number of deliveries, and professional situation were collected.

The free program G* Power (G* Power 3.1.9.4. version) [28,29] was used to determine the sample size; for a target power of 0.90, a β of 10%, an accuracy of 3%, and an α of 0.05, we needed a sample of 92 participants, which was increased by up to 15% to counter sample losses. Finally, the sample was composed of 104 participants.

The Google Forms platform was chosen for the completion and distribution because of its flexible and cheap methods of obtaining information [30]. With the prior informed consent of each participant, in order to obtain informed consent, the study conditions and compliance with the confidentiality of the subject’s personal data were drafted in a first qualifying question; only those who accepted the conditions could continue to conduct the survey. Otherwise, the questionnaire would be closed.

This project was approved by the Ethics Committee of the University of Malaga (153-2022-H). All the participants gave their consent to be part of the study before answering the APFQ.

### 2.2. Translation and Cultural Adaptation

We followed the suggestions of Wild et al. [31], who strongly recommended using the translation-back translation method with bilingual translators and then performing a thorough analysis of the new version to identify discrepancies and to verify that the questionnaire would be clearly understood by study participants.

First, a bilingual English-to-Spanish translator (native Spanish) performed a translation of the APFQ into Spanish. A consensus was reached between the research team and the translator, with which the first version was achieved. Afterward, a retranslation from Spanish to English was carried out by a bilingual English-to-Spanish translator (native English).

With the first version of the questionnaire and the retranslated version, a committee of experts agreed that there was a semantic similarity, so the version of the APFQ in Spanish was considered valid. It was first administered to a group of 10 women with characteristics similar to those of the participants in the validation process as a pilot test to detect difficulties or issues to improve. Once the pilot test was completed, the questionnaire was administered to the final sample of the study to carry out verifications regarding its reliability and validity. The questionnaire was administered online, and the participants had to complete, apart from the APFQ, the Pelvic Floor Dysfunctions Questionnaire-short version (PFDI-20) [21] and the Women’s Sexual Function Questionnaire (FMS) [32], all of them through the Google Forms platform, to ensure their anonymity and the protection of their data. The online format was chosen because of its speed in obtaining information and because it is a useful and free method [30].

After this first round of questionnaires, the reliability of the questionnaire, as well as the criterion validity and construct validity, which was verified to ensure that the composition of its dimensions corresponds to that of the original scale.

To analyze the temporal stability, a test-retest test was performed. To do this, after 15 days, the participants were invited to complete the APFQ again. In order to link the 2 results that each participant gave for this questionnaire, on the first occasion, instructions were provided for the generation of a code that had to be entered in the second answer of the APFQ.

### 2.3. Data Analysis

We used SPSS version 26.00 for Windows (SPSS Inc., Chicago, IL, USA) for all of the statistical analyses.

First, a descriptive study of the variables that characterize the sample was carried out, as well as the scores obtained in the questionnaire and each of the partial dimensions that comprise it.

Next, an analysis of the psychometric characteristics of the Spanish version of the APPQ was carried out. Consequently, Cronbach’s alpha coefficient was used to estimate the internal consistency of the global questionnaire, considering anything ranging from 0.70 to 0.95 as optimal [33]. We also calculated whether any of the items were deleted in order to check whether all of the items contributed to an optimum alpha coefficient. The correlations between items were calculated with the Pearson correlation coefficient to assess consistency, as well as the item–total scale correlation to prove each item’s correlations with the global questionnaire. Adequate internal consistency is considered to be found when the α coefficient is equal to or higher than 0.70 [34]. We interpreted these coefficients according to Cohen [35], as follows: a low correlation for a coefficient value of 0.1, a moderate correlation for a value of 0.3, and a high correlation for a value of 0.5.

Temporal stability was analyzed using a correlation analysis between both measurements (Spearman’s rho).

To analyze the criterion validity, the correlation between the APFQ and the PFDI-20 [21] and FMS [32] questionnaires was verified with Spearman’s rho coefficient.

Finally, we performed analyzed the construct’s validity to ensure that the composition of the experimental scale dimensions corresponded to that of the original scale. An exploratory factor analysis with both the principal components extraction method and the varimax rotation method was performed. These methods are used to identify underlying constructs or factors that can explain the correlations among a set of items and summarize a large number of items with a smaller number of derived items, called factors [36]. The inclusion criterion for considering a factor valid was an eigenvalue over 1 [37].

Before performing the exploratory factor analysis, we needed to prove its suitability. We did so by using a correlation matrix, the Kaiser–Meyer–Olkin (KMO) measurement of sampling adequacy, and Bartlett’s test of sphericity. The KMO measurement of sampling adequacy tests whether the partial correlations among items are small. A KMO value of greater than 0.7 indicates a strong correlation, meaning that factor analysis should be a useful technique.

Bartlett’s test of sphericity assesses whether the correlation matrix is an identity matrix, indicating that the factor model was inappropriate [36].

Finally, we calculate the standard error of measurement (SEM), which is calculated using the formula SEM = DT√(1 − α). Measurement error is the systematic and random error in a patient’s score that is not attributable to actual changes in the construct to be measured [34].

## 3. Results

After the pilot test, the panel of experts finally agreed on the final version of the APFQ for validation. The final sample of the study included 104 participants after excluding people who did not provide informed consent or who completed the questionnaire incorrectly (*n* = 16). The participants were recruited through consultations with the researchers and consultations with collaborating physiotherapists who volunteered to disseminate the questionnaire and recruit participants among their patients.

Of the women participating in the study, the age mean was 37.7 (SD 9.27); all of them had given birth, and half had given birth twice (*n* = 50), both vaginally and by cesarean.

### 3.1. Questionnaire Scores

We obtained a total questionnaire score of 12.44 (SD ± 6.9) for *n* = 104. Based on the dimensions, an average of 6.67 (SD ± 4.9) was obtained for bladder function; 3.87 (SD ± 3.1) for bowel function; 0.53 (SD ± 1.3) for prolapse; and 1.35 (SD ± 2.1) for sexual function.

### 3.2. Reliability Analysis

To analyze the questionnaire’s reliability, the internal consistency was evaluated using Cronbach’s alpha [38]. This coefficient ranges between 0 and 1, with values over 0.60 indicating acceptable reliability and values over 0.7 indicating high reliability. A Cronbach’s alpha coefficient of 0.795 was obtained for the sum of the final questionnaire, indicating high reliability.

Cronbach’s alpha was subsequently calculated for each of the dimensions of the questionnaire separately, obtaining the following results: 0.864 for bladder function; 0.796 for bowel function; 0.851 for prolapse; and 0.418 for sexual function (Table 1).

Similarly, the item–scale correlation was calculated for each dimension that made up the questionnaire. The items in which elimination increased Cronbach’s Alpha are shown in Table 2. Of those, items 3–12 belong to the dimension “bladder function,” and items 17–24 belong to “bowel function.”

For the temporal stability analysis, 96 answers were recruited, so eight participants dropped out of the study due to a lack of interest. The result of the correlation test was a Spearman’s coefficient value of 0.778 (*p* ≤ 0.000).

### 3.3. Criterion Validity Analysis

A criterion validity analysis of the bladder, bowel, and POP dimensions of the APFQ and PDFI-20 was performed. To do this, Spearman’s rho was calculated, obtaining a variable correlation according to the dimension: 0.704 for the bladder function dimension (*p* < 0.000), 0.462 for the intestinal function dimension (*p* < 0.000), and 0.337 for the prolapse dimension (*p* < 0.000; Table 3).

Regarding sexual function, we analyzed the criterion validity compared with the FSM, and a correlation of −0.401 (*p* < 0.000) was obtained as a result (Table 4).

### 3.4. Construct Validity Analysis

To evaluate the construct validity, a factorial analysis was carried out using the extraction method and the Varimax rotation method. Factor analysis is used to simplify information in a matrix and make it easier to interpret.

The KMO sampling adequacy tests and Bartlett’s sphericity test were used. The KMO sample adequacy test resulted in 0.718. Bartlett’s sphericity test resulted in a *p* < 0.000.

In the extraction of principal components, there were eight dimensions that explained 70% of the variance, particularly two of them with 19.73% and 15.27%; see the sedimentation graph in Figure 1.

The rotated matrix (Varimax rotation) shows that dimensions 1, 4, 7, and 8 are correlated with urinary function items. Dimensions 2 and 5 include items related to bowel function, and dimension 3 includes items related to prolapse symptoms.

We calculated the SEM using the formula SEM = DT √(1 − α), and we obtained a value of 3.21. Given the average score of 12.64 at the initial test measurement, there are 95 out of 100 chances that the individual’s true score would fall between 4.36 and 20.92.

## 4. Discussion

In this study, we presented a cross-cultural adaptation of the APFQ questionnaire to the Spanish population. It has high reliability, according to our results, as with the versions of the questionnaire used for other populations [39,40,41]. In addition, the measurements remained stable over time, making it a good instrument to reflect changes in the evolution of subjects. There is no other questionnaire in Spanish that includes the measurement of every pelvic floor disorder. That was the reason the APFQ was considered a useful tool to assess pelvic floor functions in daily clinical practice. The APFQ also assesses severity and bothersomeness, which may be important information for clinical practice [24].

In the analysis of the dimensions in isolation, they are reliable, except for the sexual function dimension; however, by removing item 37, Cronbach’s alpha increases notably (0.67). One of the reasons why its reliability is lower could be the two different versions of this dimension that arise depending on responses to the first item (question 33), which is not scored. In this item, the patient is asked whether or not she is sexually active. If the selected option is “not sexually active,” only two more items are answered, 34 and 42, the latter being the only one that provides a score within this dimension.

However, if the patient is sexually active, she answers 10 questions. Another reason could be the variability of the types of responses within this dimension since not all of them are Likert types with four options. On the one hand, item 35 has two possible response options. On the other hand, item 34 has 6 possible response options, with a total score of 18 points for 3 of them, the rest being valued at 0 points.

As for the rest of the dimensions, in bladder function, the suppression of items 3, 8, 9, and 12 increased the reliability of this dimension separately. The same happens with questions 17, 21, 22, and 24 of the intestinal function dimension.

Therefore we may suggest using a shorter version of the questionnaire, eliminating the items which worsen reliability. This could help make a faster assessment with an absolutely reliable and valid instrument.

Regarding the criterion validity, comparing the APFQ [26] and PFDI-20 [21], a strong correlation was found for bladder function, a moderate correlation was found for bowel function, and a weak correlation was found for prolapse symptoms. Therefore, it can be concluded that these dimensions measure the object of study in a similar way. The difference between these two questionnaires is the way the APFQ obtains more information; for example, in the urinary function dimension, data are obtained regarding recurrent urinary infections; limitations in fluid intake to avoid losses; if the patient uses or does not compress for leaks; if there is pain during urination; how it affects daily life; etc. This information is of great value in knowing the pathology’s effects on the patient’s life; in addition, it is information that can be used to offer education to the patient about her pathology and habits. Likewise, the advantage of the APFQ over other pelvic floor questionnaires validated in Spanish is that it is the only one that introduces so many pelvic floor pathology dimensions in the same questionnaire.

In the factor analysis, we recorded eight factors with eigenvalues of greater than 1.00, although two of these eight accounted for most of the variance. The rotated-component matrix is useful for determining whether each variable has substantial loading on 1 factor. The size of loading that can be called substantial is a subject about which there are many divergent opinions, and the loading depends on the sample size and the total number of variables. The larger the sample size or, the larger the number of variables, the lower the loading needs to be. A solid factor is considered to be well-defined when more than 1 variable has a loading of greater than 0.5 [42], and that criterion was the one that we followed in the present study.

We obtained 4 factors related to the bladder function domain, factors 1, 5, 6 and 7, which mainly represent the variables related to daily self-care strategies, quality of life, mictional behavior, and severity of the symptoms; factor 2 represents six variables related to the bowel function; factor 3 gathers together four variables related to the presence and severity of prolapse; and the fourth factor includes five variables related to the sexual function.

Factor 8, however, although its eigenvalue is over one, represents three variables that are not connected among them.

The questionnaire has also been shown to have good temporal stability a high correlation coefficient was obtained. In order to avoid the effect of memory in the second answer, a two-week lapse between both measurements was considered enough.

As the main implication for clinical practice, we believe that this cross-cultural adaptation of the APFQ to Spanish can be very useful for all professionals who work with patients with pelvic floor dysfunctions since it integrates the most representative spheres of these dysfunctions into a single tool, providing a global vision of the patient’s evolution.

Its transformation into a self-administered questionnaire makes it faster as a clinical tool since it is not a therapist-dependent diagnostic test. In addition to not involving costs, it is a tool that can be used online, a platform that is increasingly being used in the healthcare area [43,44].

One of the limitations that we encountered when carrying out this study was that, despite providing us with information about the questionnaire functionality, it was delivered online, meaning that there was no professional accompanying the respondent in filling it out; therefore, the circumstances in which the questionnaire was completed are unknown. Another limitation is the small available sample of subjects (*n* = 104) we were able to access during the study.

## 5. Conclusions

The Spanish version of the APFQ can be considered a reliable and valid questionnaire, useful to identify pelvic floor disorders and assess the effectiveness of any therapeutical procedure. Its administration via an online platform may be a great advantage for the patients.

Reliability can be improved by eliminating Items 3, 7–9, 12, 17, and 21–24, so a shorter version of the questionnaire should be validated and considered for clinical practice use.

## Figures and Tables

**Figure 1 jpm-13-00940-f001:**
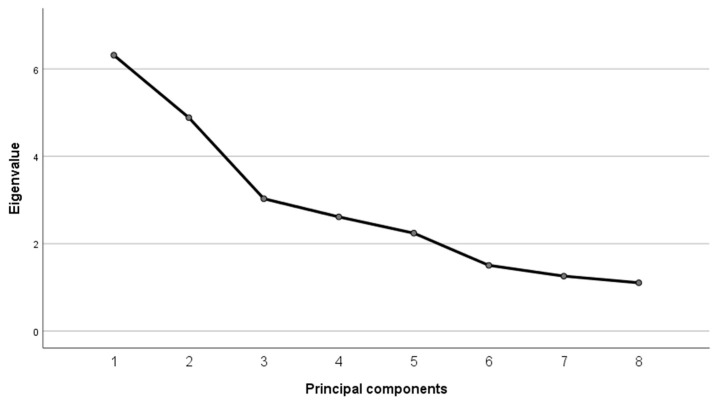
Sedimentation Graph: eight factors with eigenvalues over one.

**Table 1 jpm-13-00940-t001:** Total scores and α values for the APFQ as a whole and for each domain.

Questionnaire Domains *n* = 104	Mean	SD	Cronbach α
Complete 42 items	12.44	±6.9	0.795
Bladder function domain	6.67	±4.9	0.864
Bowel function domain	3.87	±3.1	0.796
Prolapse domain	0.53	±1.3	0.851
Sexual function domain	1.35	±2.1	0.418

**Table 2 jpm-13-00940-t002:** Table of Items—total scale correlations. Only items in which elimination increased Cronbach’s Alpha value are shown.

	Scale Mean If Item Was Deleted	Scale Variance If Item Was Deleted	Corrected Item—Total Correlation	Squared Multiple Correlation	Cronbach’s Alpha Value If Item Was Deleted
Item 3	9.99	55,000	0.304	0.311	0.866
Item 7	9.34	50,905	0.343	0.634	0.864
Item 8	9.38	51,598	0.296	0.703	0.865
Item 9	9.46	51,804	0.263	0.669	0.867
Item 12	9.66	53,526	0.213	0.539	0.867
Item 17	5.60	21,311	0.225	0.227	0.797
Item 21	5.00	20,932	0.077	0.439	0.824
Item 22	5.28	20,630	0.207	0.416	0.802
Item 23	5.70	21,337	0.281	0.378	0.795
Item 24	5.76	21,913	0.185	0.309	0.799

**Table 3 jpm-13-00940-t003:** Criterion validity of the bladder, bowel, and POP dimensions of the APFQ and PDFI-20.

		Prolapse Function PDFI-20	Bowel Function PDFI-20	Bladder Function PDFI-20
Spearman’s Rho	Bladder function APFQ	0.704 0.000 N = 104		
	Bowel function APFQ	0.462 0.000 N = 104		
	Prolapse function APFQ	0.337 0.000 N = 104		

**Table 4 jpm-13-00940-t004:** Criterion validity of the sexual dimension of the APFQ and FSM.

		Sexual Function FSM
Spearman’s Rho	Sexual function APFQ	−0.401 0.000 N = 104

## Data Availability

The data presented in this study are available on request from the corresponding author. The data are not publicly available due to legal privacy restrictions.

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
