# Peer review of "Spanish Cross-Cultural Adaptation of the Australian Pelvic Floor Questionnaire"

_jpm, 2023, doi:10.3390/jpm13060940_

Round 1
Reviewer 1 Report
In this study, the authors carry out the translation and cross-cultural adaptation of the Australian Pelvic Floor Questionnaire (APFQ) into Spanish and they evaluate its psychometric properties of validity and reliability in the Spanish population. They revealed that the Spanish version of the APFQ is a reliable and valid tool to assess symptoms and impacts on quality of life due to pelvic floor dysfunctions in the Spanish population.
First, I would like to congratulate to authors on their work. It is well structured, shows a mastery of the area and their writing allows the reader to understand the research presented.
Therefore, I am going to recommend this publication after minor revisions that I will advise the authors to consider:
Methods:
- Please explain how you obtained the informed consent in the anonymous web survey.
- Please explain why didn’t you include the elderly population and those without diagnosis and/or symptoms.
- Did you account for other conditions/diseases/drugs that might have influenced the results?
Results:
- I would recommend adding correlation results with other questionnaires (section 3.3) into a new table.
- In Table 1, please define the abbreviations and explain each dimension again.
- In Figure 1, please add axes’ names and specify which 8 factors.
Minor and technical:
- Line 71, replace APPQ with APFQ.
- Line 90, shouldn’t alpha be 0.05?
- I can not access supplementary files.
Author Response
Dear reviewer,
The authors are all very grateful for your kind assessment and considerations.
Please, read the answers to your suggestions and comments below:
Methods:
-Please explain how you obtained the informed consent in the anonymous web survey.
Here is the text we have added in lines 105-109:
In order to obtain informed consent, the study conditions and compliance with the confidentiality of the subject’s personal data were drafted in a first qualifying question, only those who accepted the conditions could continue to conduct the survey; otherwise the questionnaire would be closed.
- Please explain why didn’t you include the elderly population and those without diagnosis and/or symptoms.
Here is the text we have added in lines 82-88:
The participation of persons over 65 years of age was rejected to include only those young adults who suffered some kind of sexual dysfunction without interfering with other pathologies; in the same way it was only considered to include people with some symptom to be able to check the state of their dysfunction through the questionnaire and thus be able to compare it with other instruments already validated in order to ensure the validity of the questionnaire.
- Did you account for other conditions/diseases/drugs that might have influenced the results?
The complete medical history of each subject was not taken into account; only these subjects had to go through consultation to seek treatment for pelvic floor dysfunction.
Results:
- I would recommend adding correlation results with other questionnaires (section 3.3) into a new table.
We added a new table showing the correlations results, Table 2, in lines 225-231, and in Table 3, in line 235.
- In Table 1, please define the abbreviations and explain each dimension again.
We have defined the abbreviations and explained which items belong to each dimension in lines 210-213.
- In Figure 1, please add axes’ names and specify which 8 factors.
We have added the axe´s names and the eight factors can now be easily identified in the graphic (Figure 1).
Minor and technical:
- Line 71, replace APPQ with APFQ.
We made the change.
- Line 90, shouldn’t alpha be 0.05?
You are so right, sorry for the mistake. It has been corrected.
- I can not access supplementary files.
We will provide the original and adapted versions of the questionnaire as supplementary files. So far, there was no other supplementary file.
Reviewer 2 Report
Dear author's
I was pleased to review your article and I have the following comments>:
The subject is interesting. Please explain the novelty of this study.
The validation of this questionnaire in Spain is a preliminary for a new study?
Please tell as why do you chose the 15 days to repeat the questionnaire?
Please reformulate the conclusion because is nor clear. The conclusion highlight the principal results of the study.
English and punctuation edits are necessary for your manuscript.
Author Response
Reviewer 2:
I was pleased to review your article and I have the following comments>:
The subject is interesting. Please explain the novelty of this study.
The main reason why we have performed a transcultural validation of this questionnaire was to unify in a single diagnostic tool the main areas of dysfunctions that encompass the pelvic floor sphere. The items of this questionnaire are simplea and easy to be answered, and qualitatively assesses the magnitude of the problem.
The validation of this questionnaire in Spain is a preliminary for a new study?
The validation of the questionnaire is the basis for future research, since our specailty is the management of pelvic floor disorders. Now, the immediate first application we need this questionnaire for, is to assess the presence and magnitude of pelvic floor disorders in Spanish female athletes, in order to carry out a clinical trial in which the participants will go under a tele-rehabilitation program.
Please tell as why do you chose the 15 days to repeat the questionnaire?
Citing Dutil et al. (Test-retest reliability of a measure of Independence in everyday activities: The ADL profile. 2017. Occupational Therapy International. doi: 10.1155/2017/3014579), “some studies have shown that though the optimal time-interval between testing will vary depending on the construct being measured, on the stability of the construct over time and on the target population, the target time of 2 weeks is the most frequently recommended interval [15 Streiner D. L., Norman G. R., Cairney J. Health Measurement Scales. A Practical Guide to Their Development and Use. 4th. New York, NY, USA: Oxford University Press; 2014. [CrossRef] [Google Scholar]]”.
Please reformulate the conclusion because is nor clear. The conclusion highlight the principal results of the study.
We have added a Conclusions section and reformulated the conclusion in lines 296-304, as folows:
The Spanish version of the APFQ can be considered as a reliable and valid questionnaire, useful to identify pelvic floor disorders and assess the effectiveness of any therapeutical procedure. Its administration via an online plaftform may be a great advantage for the patients.
Reliability can be improved by eliminating Items 3, 7-9, 12, 17, 21-24, so a shorter version of the questionnaire, should be validated and considered for clinical practice use.
English and punctuation edits are necessary for your manuscript.
Thank you, we already sent the manuscript to the MDPI editing service. We will contact them again and ask for a second review.
Reviewer 3 Report
The aim of this study was to carry out the translation and cross-cultural adaptation of the Australian Pelvic Floor Questionnaire (APFQ) into Spanish and the evaluation of its 12 psychometric properties of validity and reliability in the Spanish population. A pilot test in 10 women were then conducted. In the introduction the main information on the topic are well reported. Although, authors should add few data regarding the impact of lower urinary tract symptoms on QoL, these two articles can help (10.3390/jcm11216572 and 10.3390/jcm11195639) I have nothing to say on the methodology. Results are well written and are well resumed in tables and graphs. In the discussion the results of the study are well reported and analysed. Limitations of the study are clear and correctly reported. The conclusion chapter should be added. A minor revision of the English language is in order.
Author Response
Dear reviewer,
The authors are all very grateful for your kind assessment and considerations.
Please, read the answers to your suggestions and comments below:
The conclusion chapter should be added.
We have added a Conclusions section and reformulated the conclusion in lines 296-304, as folows:
The Spanish version of the APFQ can be considered as a reliable and valid questionnaire, useful to identify pelvic floor disorders and assess the effectiveness of any therapeutical procedure. Its administration via an online plaftform may be a great advantage for the patients.
Reliability can be improved by eliminating Items 3, 7-9, 12, 17, 21-24, so a shorter version of the questionnaire, should be validated and considered for clinical practice use.
Authors should add few data regarding the impact of lower urinary tract symptoms on QoL, these two articles can help (10.3390/jcm11216572 and 10.3390/jcm11195639)
Thank you very much for the references, we have included them and some more data in lines 38-41. Unfortunalely, we couldn´t add more information since the manuscript already has a high number of words:
These symptoms are even more concerning in neurological patients, with a prevalence of 80% with pelvic floor disorders(3). These conditions have a negative impact on quality of life and in different areas of the patients´ life, such as social, psychological, work or sexual sphere(4,5,6).
A minor revision of the English language is in order.
Thank you, we already sent the manuscript to the MDPI editing service. We will contact them again and ask for a second review.
Round 2
Reviewer 2 Report
Dear author’s
Thank you for your response.
Reviewer 3 Report
The authors revised the manuscript. I have nothing to add.